# Micro-Injection Molding of Diffractive Structured Surfaces

**Ann-Katrin Boinski [1,2], Barnabas Adam [1,2,*], Arne Vogelsang [3], Lars Schönemann [1,2]****, Oltmann Riemer [1,2]** **and Bernhard Karpuschewski [1,2]**

1   Leibniz Institute for Materials Engineering IWT, LFM, Badgasteiner Str. 2, 28359 Bremen, Germany; boinski@iwt.uni-bremen.de (A.-K.B.); schoenemann@iwt.uni-bremen.de (L.S.); riemer@iwt.uni-bremen.de (O.R.); karpuschewski@iwt-bremen.de (B.K.)
2   MAPEX Center for Materials and Processes, University of Bremen, 28359 Bremen, Germany
3   POLYoptics GmbH, Kölner Straße 101–107, 50389 Wesseling, Germany; arne.vogelsang@polyoptics.de
*   Correspondence: adam@iwt.uni-bremen.de; Tel.: +49-421-218-511-70

**Abstract:** In recent years, the use of highly functional optical elements has made its way into our everyday life. Its applications range from use in utility items such as cell phone cameras up to security elements on banknotes or production goods. For this purpose, the Leibniz Institute for Materials Engineering (IWT) has been developing a cutting process for the fast and cost-effective production of hologram-based diffractive optical elements. In contrast to established non-mechanical manufacturing processes, such as laser lithography or chemical etching, which are able to produce optics in large quantities and with high accuracy, the diamond turning approach is extending these properties by offering several degrees of freedom. This allows for an almost unlimited geometric complexity and a structured area of considerable size (several tenth square millimeters), achieved in a single process step. In order to introduce diffractive security features to the mass market and to actual production goods, a high-performance replication process is required as the consecutive development step. Micro injection molding represents a feasible and promising option here. In particular, diamond machining enables the integration of safety features directly into the mold insert. Not only does this make additional assembly obsolete, but the safety feature can also be placed inconspicuously in the final product. In this paper, the potential of micro-injection molding as a replication process for diffractive structured surfaces will be investigated and demonstrated. Furthermore, the optical functionality after replication will be verified and evaluated.

**Keywords:** diamond machining; diffractive optics; security features; micro-injection molding; nano fast tool servo

## 1. Introduction

Counterfeiting is a widespread problem no matter whether the financial sector, general production goods or medical technology is concerned. For this reason, the demand for appropriate protection mechanisms has remained consistently high for several decades. The conventional production of diffractive optical elements is mainly covered by lithographic processes [1,2]. Nevertheless, due to long production times, this process is very expensive when it comes to mass production and the structuring of large areas. In order to meet the high demands on geometrical quality and surface finish, and furthermore, to achieve an extension of the producible geometries, mechanical diamond machining processes have been developed, which allow the fast and cost effective machining of diffractive structured surfaces [3]. The processes for the machining of diffractive structured surface are numerous and range from micro milling [4] to diamond micro chiseling [5] and diamond turning [6].

In order to further expand the flexibility and degrees of freedom of the machining process, ultra-precision machining processes can be combined with external additional axes like piezo-driven fast tool servos (FTS). The potential applications of these high-precision

and highly dynamic additional axes are numerous. Yu et.al, for example, are using a FTS for further increasing the accuracy of the machining process by directly compensating deviations from the tool path and thus compensating machining errors [7]. Another possibility is the combination of fast and slow tool axes for the machining of hierarchical structures, such as structured freeform surfaces, in which a long-wave microstructure is overlaid by a diffractive nanostructure [8]. A similar approach was followed by the Leibniz-Institute for Materials Engineering (IWT), which has developed a manufacturing process which is combining a conventional face-turning diamond cutting process with a high-frequency nano fast tool servo (nFTS) and the application of thin diamond tools of rectangular shape. This process is capable of machining diffractive optical elements (DOE) of considerable size, fulfilling highest demands on the geometric quality and surface finish and all these within a single process step with short production time and therefore low financial expenses [9].

Although the structuring process offers significant advantages in terms of speed, flexibility and size of the area to be structured, there are still some limitations for use in mass production. One is the structuring of polymers, which make up most of our goods today. Therefore, a subsequent replication process has to be found in order to enable a cost-effective mass production of micro-structured surfaces.

The three main molding processes are hot embossing, UV-embossing and injection molding. These processes are attractive especially for the replication of micro- and nanostructures, by offering high geometric resolution of the replicated structure, by showing high flexibility in the complexity of the replicated structure, and finally, all are very cost effective. Hot stamping and UV-embossing are particularly suitable for the structuring of thin films. However, in order to be able to integrate a safety feature directly into a three-dimensional volume, injection molding it the technology of choice [10].

Micro-injection molding can be seen as the miniaturized variant of conventional injection molding, which is characterized in particular by a relatively large surface-to-volume ratio and a very sophisticated filling behavior of the cavity [11]. For this reason, considerable effort has been invested in recent years in the investigation and optimization of the micro-injection molding of structures with high aspect ratios [12–14]. With this knowledge and using appropriate process parameters, it is possible to reproduce diffractive structures in large quantities exhibiting sub-micrometer features with dimensions between 1–10 μm range and sub-micrometer surface finishes [13,15].

## 2. Materials and Methods

### 2.1. Diamond Turning of Diffractive Structured Surfaces

The machining of the mold insert is conducted on an ultraprecision machine tool Nanotech 350 FG, combining a conventional face turning process with an additional high frequency piezo actor. The so-called nano fast tool servo is capable of modulating the depth of cut with a working frequency of 5 kHz and nanometer precision, generating up to 2000 structure elements per revolution on the helical tool path (Figure 1a). The height of each structure element is calculated in advance and can be controlled individually with the encoder signal of the main spindle working as a trigger for the nFTS movement. The maximal deflection of the nFTS is 1000 nm with an accuracy of 4 nm. The height differences between the individual pixels thus lie within this range. In cutting direction (Figure 1b), the tool path of this structuring process can be seen. It is noticeable that when plunging into the surface, the clearance angle of the diamond tool is defining the angle of the transition between neighboring structure elements. On the other side, when decreasing the depth of cut, the speed of the nFTS is determining the angle of the edge. Due to the process kinematics, the pixel-size in this direction in decreasing with the radial distance from the workpiece center. By using narrow rectangular diamond tools and a feed rate corresponding to the width of the tool, the characteristic rectangular microstructure is applied to the surface, which can be seen in the cross-section in Figure 1c. The horizontal size of the pixels lies in the micrometer range. In this way, it is possible to structure an area of about 30 square millimeters in under an hour. In previous investigations,

different materials were examined for their structurability in terms of shape accuracy and surface finish. Nickel silver (CuNi7Zn39Pb3Mn2) has proven to be particularly suitable for this purpose. In comparison to ultrafine grained aluminum, it is characterized by a higher hardness, which makes the material appropriate for use as a molding tool [16]. All process parameters, as well as the tool configuration, are listed in Table 1.

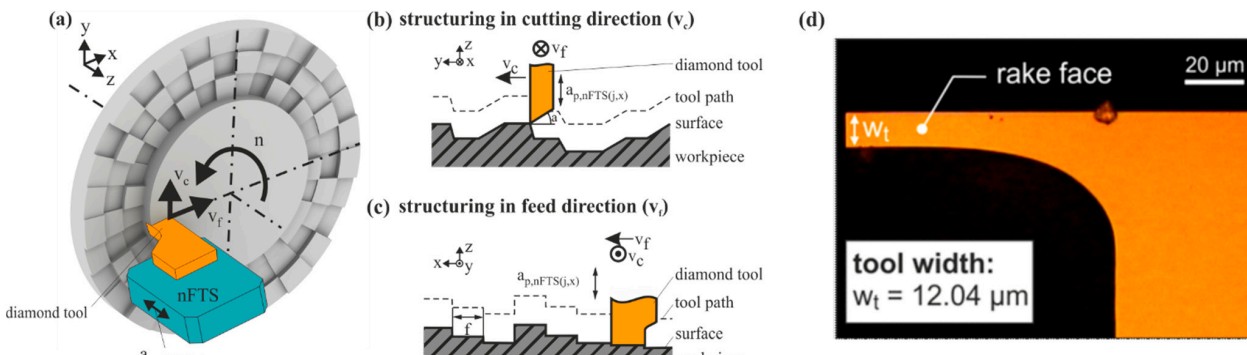

**Figure 1.** Process principle of diamond machining of diffractive structured surfaces. (**a**) Workpiece and nFTS with diamond tool, (**b**) structuring tool path in cutting direction, (**c**) structuring tool path in feed direction, (**d**) rectangular shaped diamond tools with a tool width of 12.04 μm.

**Table 1.** Process parameters for the machining of diffractive structured surfaces by diamond turning with nFTS and tool configuration.

| Machining Parameters | | Diamond Tool | |
|---|---|---|---|
| Mold material | Nickel Silver [CuNi7Zn39Pb3Mn2] | Tool material | Monocrystalline Diamond |
| Feed f | 11 μm | Tool width | 12.04 μm |
| Depth of cut $a_p$ | 2 μm | Geometry | Rectangular |
| Depth of cut nFTS $a_{p,nFTS}$ | 0 to 1000 nm | Clearance angle $\alpha$ | 20° |
| Spindle speed | 100 rpm | Rake angle $\gamma$ | 0° |
| Working frequency nFTS | 5 kHz | | |
| Accuracy nFTS | 4 nm | | |

As mentioned before, the resulting microstructure is not randomly chosen, but calculated in an algorithm in advance of the structuring process. This optical design is calculating the height of each structure element, resulting in a surface structure capable of modulating the phase of laser radiation of specific wavelength. The reflected light then creates a defined intensity distribution on a reconstruction plane [17]. To preserve the functionality on the molded part and not on the structured mold insert, the calculated structure is inverted before machining, generating the negative diffractive structure on the metal surface.

### 2.2. Injection Molding of Diffractive Structured Surfaces

Two criteria were decisive for a successful replication of the diffractive structures: The refractive index of the chosen polymer and the molding behavior, especially a high melt flow rate (MFR). This is why a cyclic olefin copolymer (COC) named Topas 5013-S04 was chosen for the replication. This material is particularly suitable for the injection molding of parts with thin injection gates, due to its good material flow properties. Also, due to the high MFR, the material flows easily into the mold insert, allowing the part to be filled completely. The refraction index must be known beforehand and is included in the optical design.

Injection molding experiments have been carried out using a micro-injection molding machine (Krauss Maffei 50, KraussMaffei Group, Munich, Germany) with maximum clamping force of 500 kN. This maximum force was also used for the experiments. The screw

had a diameter of 20 mm and can therefore fill one or two cavities without causing an ashing effect of the material. The mold design is shown in Figure 2. The mold is made for two cavities, but for the injection molding tests only one cavity (marked green) was used. The other cavity was blocked and will not be filled. Still the sprue is longer than for a one cavity design. A tunnel gate which will cut automatically when demolding the part is used. The polymer applied (Topas 5013-S04, TOPAS Advanced Polymers GmbH, Raunheim, Germany) was pre-dried at 100 °C for three hours before it was filled into the machine.

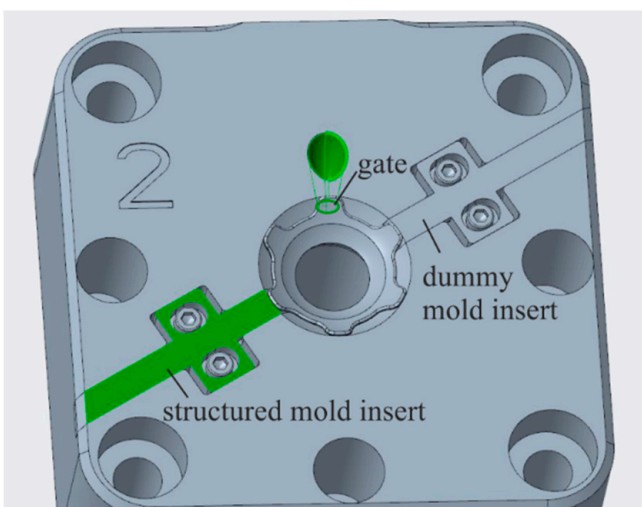

**Figure 2.** Mold design of the injection mold with mold insert.

In the isotherm process, two cooling circuits are used: one for the outer plates, which are heated to 85 °C, and a second circuit for the cavity, which was set at 145 °C. In fact, after a few passes the cavity temperature will increase even more as the material is injected at around 275 °C in the last zone of the screw. The respective parameters are initially found in filling studies and then tested for the final optical product in terms of achievable surface quality, listed in Table 2.

**Table 2.** Material properties and Process parameters for the micro-injection molding of diffractive structured surfaces.

| Material Parameters of COC, Topas 5013-S04 | | |
|---|---|---|
| **Property** | **Test Method** | **Value** |
| Refractive index (580 nm, 25 °C) | ISO 489 | 1.533 |
| Density | ISO 1133 | 1020 kg/m$^3$ |
| Melt flow rate (MFR) (260 °C, 2.16 kg) | ISO 1133 | 43 g/10 min |
| **Molding Parameters** | | |
| Molding machine | | Krauss Maffei 50 |
| Profile | | 1 cm$^3$/s |
| Plasticizing volume | | 3.3 cm$^3$/cavity |
| Residual mass cushion | | 1.75 cm$^3$/cavity |
| Specific dynamic pressure | | 100 bar |
| Maximum injection pressure | | 1463 bar |
| Reprint | | 900 bar (for 2 s); 800 bar (for 0.3 s) |
| Injection time | | 2.58 s |
| Holding time | | 2.3 s |
| Residual cooling time | | 29 s |
| Demolding | | 6.98 s |
| Total cycle time | | 46 s |

## 3. Results

### 3.1. Experimental Results—Replication of Diffractive Structured Surfaces

In order to examine the molding quality initially on simple geometries, the first replication experiments were carried out using test structures, of grooves with several changeover points, specific depth and equal width. It was found that the replication quality of the diffractive structures is very high. In experiments with rectangular test structures, the geometry and sharpness of the edges could be well reproduced. In Figure 3, a corresponding position on the mold insert, as well as on the replication, is shown. It can be seen that the structure height and structure width are accurately mapped. The intended structure width was 22 μm, the structure height of the regarded structure element was intended to be 800 nm. However, burr formation occurs during the molding. Due to the measuring method with an optical white light interferometer (WLI), however, the reason for this can also be a so-called batwing-effect, which is known to occur in the optical measurement of steep flanks. This assumption is contradicted by atomic force microscopic (AFM) measurements, which are also shown in Figure 3. On the replication, a burr formation is also visible. At this particular example, the surface roughness on a specific reference position is similar on the mold and replicated part for both measurement systems. A slight shrinking behavior of the height steps of the molded structure was also detected. At this point, the shrinkage in the vertical direction was primarily considered, as a significant deviation from the desired structure height would diminish the functionality of the surface structure. It was noticeable, that the extent of shrinkage was dependent on the height of the structure. A larger difference in structure height went along with a higher shrinkage of the replicated structure element.

The surface quality was evaluated in small sections on several structure elements and an average value was calculated based on the white light interferometric and atomic force microscopic measurements. It was found that the roughness on the mold insert lies in the optical range, i.e., about 10 nm. On the molded structure the measured roughness is slightly higher, but at 12.7 nm (AFM) or 12.4 nm (WLI) it is still close to the optical range.

In Figure 4, white light interferometric measurements of a diffractive structured mold insert and its replication is shown as well as the respective heights of the associated datapoints of the previously calculated structuring dataset. Again, the measurements have been taken at the same position on the mold and replicated part. In the cross sections of these structures (Figure 4c–e), reference points are marking equal positions. Again, the replication quality is very high, although the transitions between two structural elements are not as sharp as in the test structures shown before. Also, burr formation occurs quite frequently on the replicated structures, which cannot be determined on the mold insert. However, a strong indication that the burrs are seen due to an optical batwing-effect of the measuring method can be seen in Figure 4d. Here, a deep gap between two structure elements is visible, which changes into a high burr. This deep gap cannot be explained by an incorrect detachment of the replication from the mold insert. Furthermore, the measuring method is theoretically not able to measure such deep gaps and steep flanks, which again indicates an optical effect rather than the actual presence of these phenomena.

Due to the small size of the structure elements of a few square micrometers, it was not possible to evaluate the surface roughness in this case.

In Table 3, the structure heights of mold and replication are compared to the intended heights of the original dataset. Comparing the measured structure heights of the mold insert with the original data, a strong correlation is visible. Looking at the differences between the structure heights of the mold insert and the replicated structure (Table 3), shrinkage behavior is also noticeable here. Finally, it should be noted that the deviation of the structure height between the nominal structure and the replica with up to 20% is regular in extent and not negligible, and an influence on the optical quality, i.e., the functionality of the structures cannot be excluded. This will be evaluated in more detail in the following section.



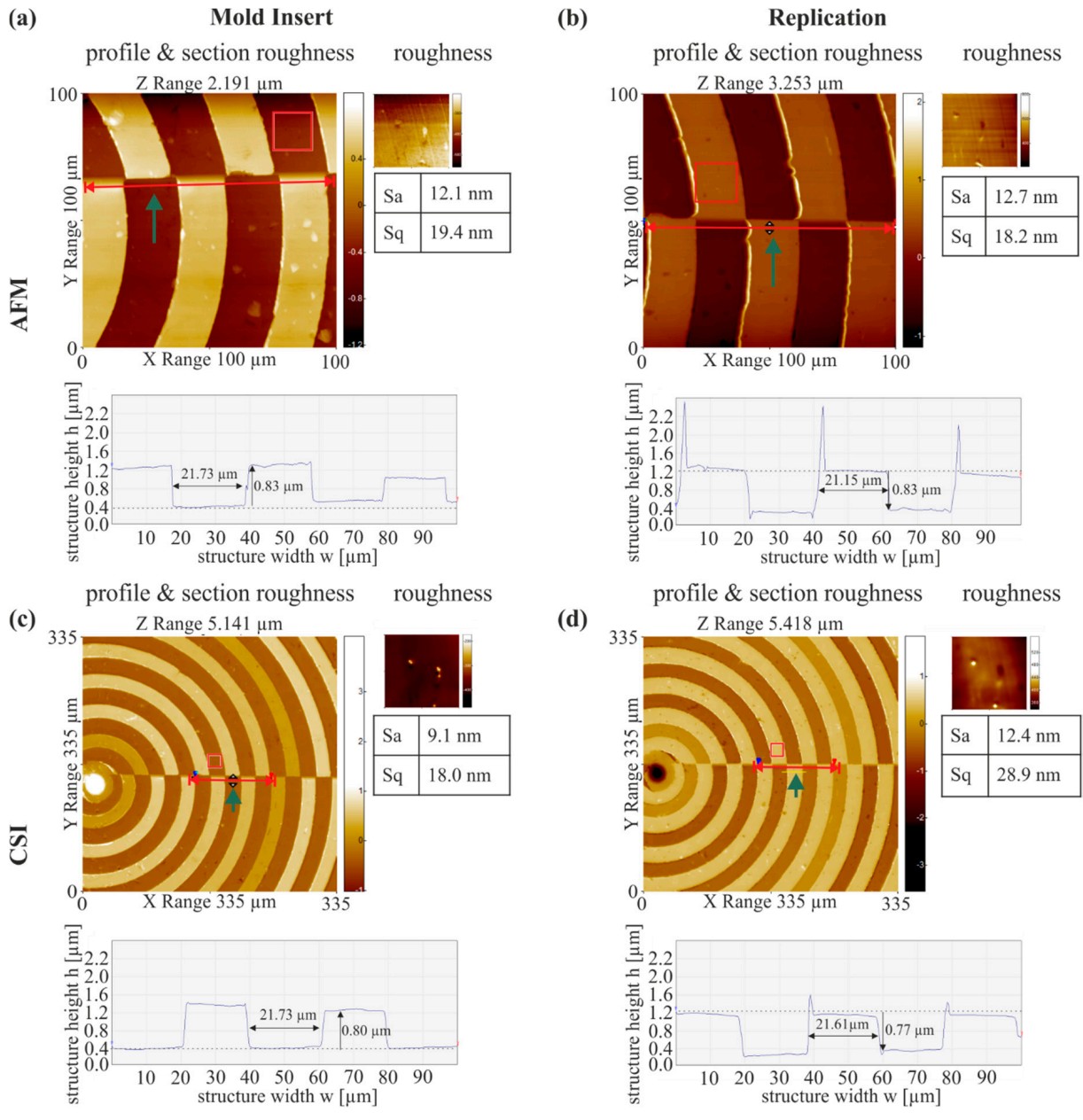

**Figure 3.** Comparison of mold (**a**,**c**) and replicated surfaces (**b**,**d**) with test structure: atomic force microscopic (AFM) measurement and cross section (**a**,**b**) and coherence scanning interferometric (CSI) measurements (**c**,**d**).

**Table 3.** Deviation of Structure heights between mold insert and replicated structure.

| Position | Height-Difference | | | Relative Deviation | | |
|---|---|---|---|---|---|---|
| | Dataset | Mold Insert | Replication | Mold to Dataset | Replication to Dataset | Mold to Replication |
| **1** | 53.0 nm | 52.1 nm | 53 nm | 98.3% | 100% | 101.7% |
| **2** | 147 nm | 148 nm | 120 nm | 100.7% | 81.6% | 81% |
| **3** | 168 nm | 172 nm | 154 nm | 102.4% | 91.7% | 90% |

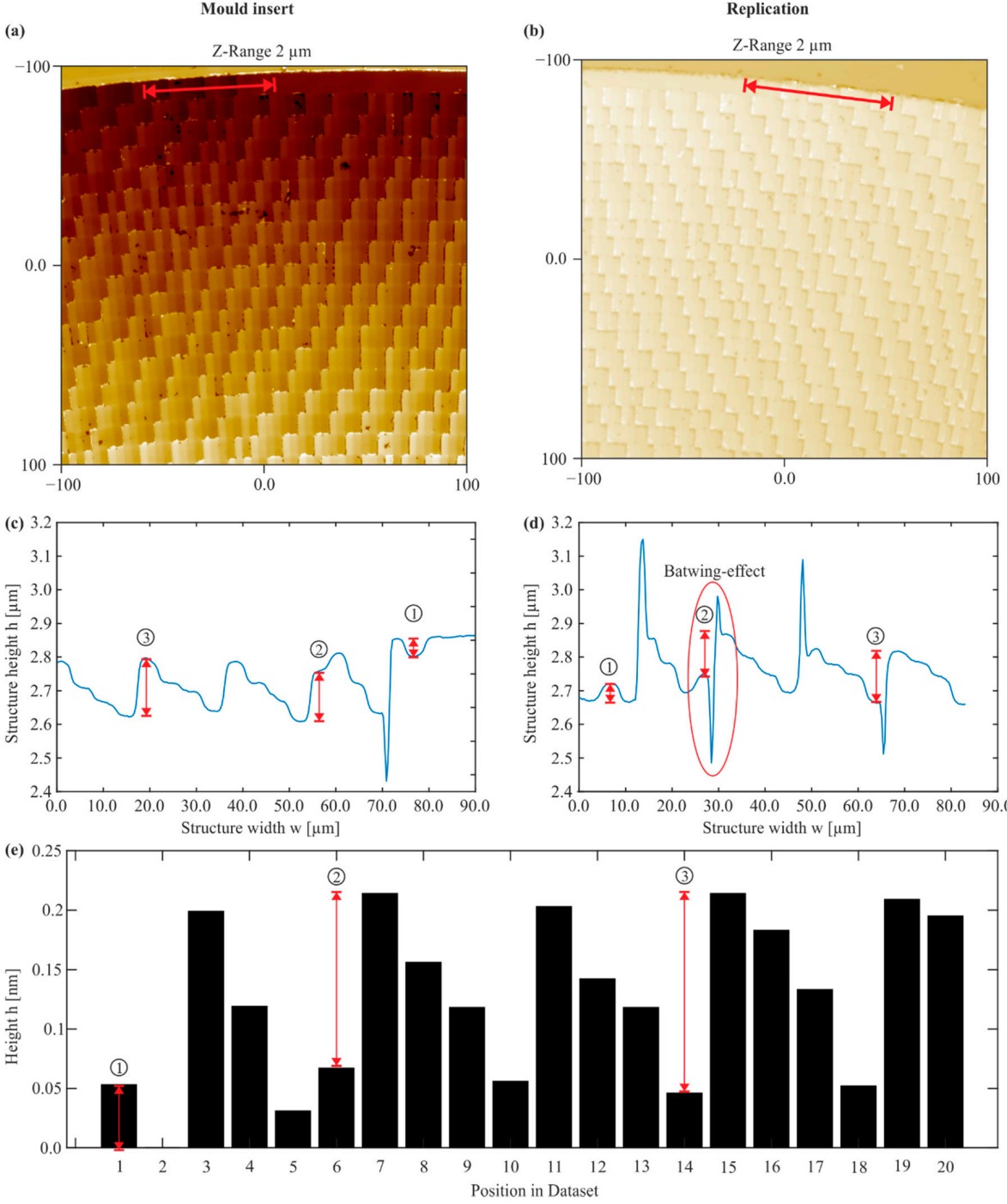

**Figure 4.** Comparison of mold and replication surface structured with a test structure, (**a**,**b**) white light interferometric measurements and (**c**,**d**) cross sections. ①, ② and ③ indicate exemplary positions for cross referencing between the measurements (**c**–**e**).

For further investigation of the structural quality of the replication, atomic force microscopic measurements were performed. Figure 5 shows an atomic force microscopic measurement and a cross section of the same replication shown in Figure 4. However, the same position is not considered. Since this is a non-optical measurement method, the batwing-effect did not occur, and the flanks of the structures can be detected by the measurement method. The results support the previously made assumption that the white light interferometric measurement of the replication shows the batwing-effect. No burr formation can be identified from the profile section. Furthermore, the structural elements

are sharply reproduced. The surface roughness in Figure 5 appears to be higher than in Figure 4, but this effect actually is attributed to the different axis scaling.

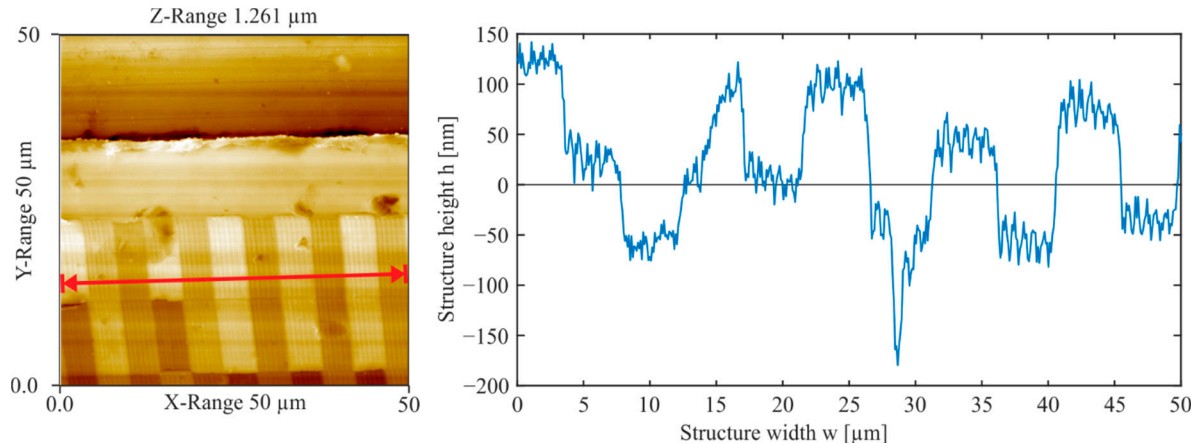

**Figure 5.** Atomic force microscopic measurement and cross section of the replication of a diffractive structured mold insert.

### 3.2. Functional Testing

The optical setup for the testing of diamond turned holograms (DTH) consisted of a laser of specific wavelength—in our case, 450 nm (blue light)—which illuminated the structured surface. The incident light is diffracted and reflected onto a reconstruction plane where the intensity distribution is visible (Figure 6a). Since the replication material is a transparent COC, the structure was sputtered with a thin gold layer first, to increase the reflectivity of the surface.

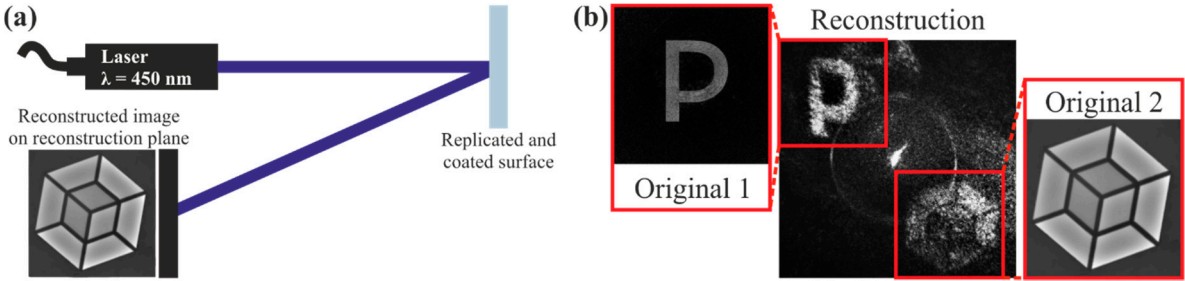

**Figure 6.** (**a**) Setup for the optical testing and (**b**) reconstruction of coated replicated surface.

Figure 6b shows the reconstructed intensity distribution after illumination with blue light. In this case, two different patterns were coded on the surface to test the limits of information density. The original patterns are shown to the left and right of the reconstruction. Both can be easily identified, although the sharpness of the edges was slightly reduced. The image sharpness would be even higher, if only one of the images and thus a lower information density would have been coded on the surface.

Another factor influencing the reconstruction quality is the edge sharpness and accuracy of the structure elements. As already noted, this deviates slightly from the desired structure due to edge rounding and shrinkage behavior. However, since the intensity distribution still can be clearly identified, this effect can be neglected for the moment.

The conformity of the original to the reconstructed image was evaluated pixel-wise for the two individual logos with the help of MATLAB. The rating was based on the matching rate of equal pixels as a percentage. In order to compare the individual images, they first had to be aligned to each other. To do this, the operator had to manually select characteristic points on both images, which were then used to superimpose the images. Figure 7 shows the resulting overlay of both logos, where black points are matching points while white

points mean the images are differing in this point. The specified conformity rate was calculated as the average of 5 evaluations in each case. It can be seen that for the Logo in Figure 7a, original and reconstructed image are agreeing fairy well (Conformity = 84.3%), while those shown in Figure 7b differed further from each other with a conformity rate of 66.6%.

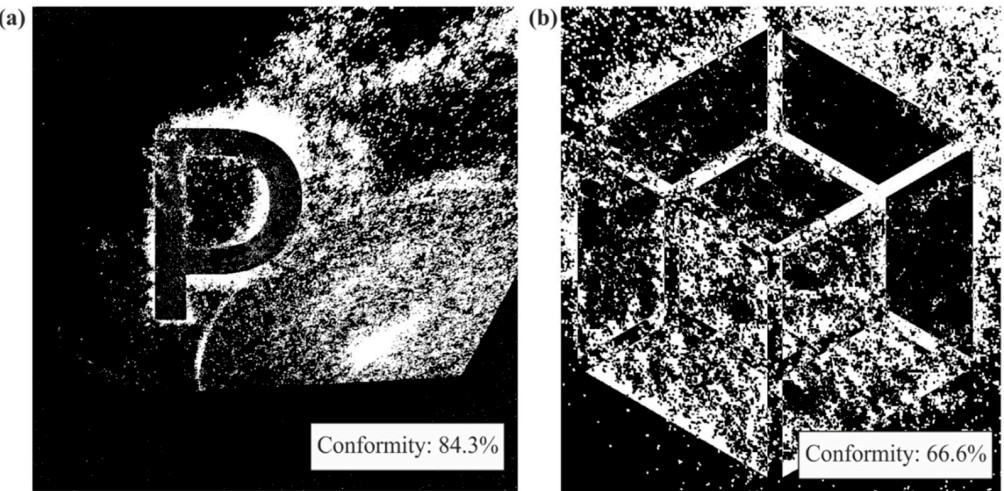

**Figure 7.** Conformity of original and reconstructed images.

## 4. Discussion

In this paper, the replication of diffractive structured surfaces by micro-injection molding has been shown. It was determined that the molding of such microstructures in a micro-injection molding process is possible in high replication quality. The minor deviations between intended structure (on the mold insert) and replicated structure were limited to a slight edge rounding and low shrinkage and occurred both in the previous test structures as well as in the subsequent diffractive microstructures.

Furthermore, the optical functionality of the replicated structure was demonstrated as a diffractive security hologram. For this purpose, an optical setup especially designed for this application was presented. Here it was possible to encode two different images on the structured surface. By illuminating the coated surface, both images were reflected as intensity distributions and made visible on a reconstruction plane. Both images could be identified with the naked eye.

To summarize, it was found that micro-injection molding is particularly suitable for the replication of diffractive structured surfaces with structures in micrometer range in lateral direction and structure heights in nanometer range. Distortions may occur but only disturb the actual functionality to a small extent.

Future investigations will focus on further improvement of the replication quality by adjusting the replication parameters and thus increasing the reconstruction accuracy of the intensity distributions. Another interesting aspect is the investigation of the wear behavior of the mold inserts in large-scale replication lines, to make the technology applicable for mass production.

**Author Contributions:** A.-K.B., B.A. and A.V. designed and performed the experiments as well as analyzed the data. A.-K.B. wrote the manuscript with support from O.R., L.S. and B.A., O.R. and B.K. supervise the project and conceived the original idea. All authors have read and agreed to the published version of the manuscript.

**Funding:** This project has received funding from the European Union's Horizon 2020 research and innovation programme under grant agreement No. 767589. Further information may be found at www.prosurf-project.eu.

**Data Availability Statement:** No new data were created or analyzed in this study. Data sharing is not applicable to this article.

**Acknowledgments:** The authors would like to thank our colleagues from the Bremen Institute for Applied Beam Technology (BIAS), especially Fabian Thiemicke, for the calculation of the optical designs as well as support in the evaluation of the reconstructions.

**Conflicts of Interest:** The authors declare no conflict of interest. The funders had no role in the design of the study; in the collection, analyses, or interpretation of data; in the writing of the manuscript, or in the decision to publish the results.

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
