# Peer review of "Micro-Injection Molding of Diffractive Structured Surfaces"

_jmmp, doi:10.3390/jmmp5010012_

Round 1

Reviewer 1 Report

The manuscript has described a process chain for optical elements starting from mould from diamond turning to replication by COC injection moulding. However, similar research work has been reported for many times, for instance, Loaldi, D., Calaon, M., Quagliotti, D., Parenti, P., Annoni, M., & Tosello, G. (2018). Evaluation of injection pressure as a process fingerprint for Injection and Injection Compression Molding of micro structured optical components. Poster session presented at Euspen Special Interest Group Meeting 2018: Structured & Freeform Surfaces, Cachan, France. And it is not the first time to use diamond turning for mold machining for optical components. 

So please argue and address the novelty of your research. 

The manuscript has mentioned diamond turning, injection moulding , geometrical measurement, replication evaluation, however, none of disussion for those parts are thorough. especially in the session 3.1, where the replication fidelity is supposed to be evaluated in a more precise way. It is not sufficient to get conclusion if only 3 positions were measured. 

it is ok to name the sharp peak on the whitelight profile as "burr", but please be aware, in most cases, burr refers to the sharp part on the mold during tooling. It may leads to misunderstanding. I would prefer "spikes" to reduce confusion. 

I would suggest to address the traceability in your measurement, i.e. use AFM as reference, but use whitelight as fast solutions (apparently those spikes is a problem). I would suggest to measure larger areas in order to compare more micro/nano structures, in order to be more convincing. 

For injection moulding part and diamond machining part, no optimization is mentioned. Therefore it will be very essential to perform proper measurements. 

The functional testing part is not very clear. However, once addressed properly, or quantified method is difined, the functional testing maybe a novel part. 

Reviewer 2 Report

Overall a nice and interesting work.

The only points to be improved are:

Chapter 'Injection molding of  ..':

Add a description of the mold design, especially the design of the cavities and the runner system. Thus the filling behavior can be judget better especially regarding certain influences from the injection molding process.

It seems that you have used a variotherm injection molding process. Add a more detailed description.

line 143: you mention a shrinkage. For which dimension was it measured?

Reviewer 3 Report

This paper is about a study on making a DOE mold using machining and making a plastic replica through injection molding. The geometric shape analysis result of the final product seems to be very good. However, it is essential to modify the contents below. 1. What is the overall size of the produced result? Also, how many cavities are applied to the mold? The information on the type and location of the gate must be added, and what is the basis for these? What analysis was applied to select the shape and location of the gate? 2. It is necessary to evaluate the uniformity of the injection product. An analysis of the uniformity of the replica, including the thickness uniformity of multiple replicas, should be added. 3. The evaluation of the reconstructed image is too qualitative. Quantitative analysis is required for the resolution of the reconstructed image obtained using conventional imaging evaluation method such as knife edge method. Also, how much difference this shows from the theoretical value should be added. 4. All results are halted in the relationship between the mold insert and the replica. It is necessary to compare the final replica with the design values. 5. At the top of Figure 2, "replication" should be modified to "replica". Also, a close check of the overall grammar is required.

Round 2

Reviewer 1 Report

It is better now to have the explanation for the funtionality evaluation. 

Reviewer 3 Report

Authors modified the manuscript. This paper is now ready for publication.